# Coalition Shaping the Vaccination Landscape

**DOI:** 10.3390/vaccines10122030

**Published:** 2022-11-28

**Authors:** Adam Antczak, Aneta Nitsch-Osuch, Marcin Balcerzak, Michał Byliniak, Joanna Chorostowska-Wynimko, Karina Jahnz-Różyk, Mikołaj Konstanty, Jarosław Krzywański, Ernest Kuchar, Agnieszka Mastalerz-Migas, Marek Wójcik, Leszek Szenborn

**Affiliations:** 1Department of General and Oncological Pulmonology, Medical University of Lodz, 90-153 Lodz, Poland; 2Department of Social Medicine and Public Health, Medical University of Warsaw, 02-091 Warsaw, Poland; 3Medink, 05-500 Mysiadlo, Poland; 4Employers’ Association of Innovative Pharmaceutical Companies INFARMA, 02-670 Warsaw, Poland; 5Department of Genetics and Clinical Immunology, National Institute of Tuberculosis and Lung Diseases, 01-138 Warsaw, Poland; 6Department of Internal Medicine, Pneumonology, Allergology and Clinical Immunology, Central Clinical Hospital of the Ministry of National Defense, Military Institute of Medicine, 04-349 Warsaw, Poland; 7Polish Pharmaceutical Chamber, 00-238 Warsaw, Poland; 8National Centre for Sports Medicine, 02-091 Warsaw, Poland; 9Department of Pediatrics with Clinical Assessment Unit, Medical University of Warsaw, 02-091 Warsaw, Poland; 10Department of Family Medicine, Wroclaw Medical University, 51-141 Wroclaw, Poland; 11Association of Polish Counties, 00-901 Warsaw, Poland; 12Clinical Department of Paediatrics and Infectious Diseases, Wroclaw Medical University, 50-368 Wroclaw, Poland

**Keywords:** influenza, promotion strategies, vaccinations

## Abstract

In 2022, the National Program for Influenza Prevention coalition will have its 10th anniversary; it is one of Poland’s oldest educational initiatives. The National Program for Influenza Prevention was initiated to prevent a further decline and promote influenza prevention in the A(H1N1) post-pandemic years. In this review, we summarize the structure and operational model of the coalition and identify core functional elements that make it a key non-governmental organization involved in the prophylactics of communicable diseases. The coalition-based organization can operate in a complex environment, such as vaccinations requiring scientific, economic, social, and psychological involvement, and communications with different groups. Anchored to the history of the National Program for Influenza Prevention, we review Poland’s vaccination landscape changes from the last ten years.

## 1. Introduction

In Europe, before the COVID-19 pandemic, influenza ranked the highest in burden of disease among all communicable diseases [1]. Historically, influenza vaccination coverage in Poland remained low (far behind most European countries) [2]. In the current century, the highest vaccination rate reported was in the 2001/2002 season, when 10.57% of Poles were vaccinated. Then, a stepwise decline in the vaccination rate occurred in all age groups, which accelerated after the 2009 A(H1N1) pandemic (Figure 1). Poland was the only country that evaluated the risk of the swine flu as low and did not order a pandemic vaccine, and the Ministry of Health neglected the effectiveness of the pandemic vaccine. Despite the effects of past decisions, the message resonated in society and strengthened the reluctance toward seasonal vaccination. As a result, in the A(H1N1) post-pandemic years, the vaccination coverage decreased twice compared to an earlier time (Figure 1). Negative attitudes toward vaccination became widespread in society and among professionals [3]. Experiences from the 2009/2010 pandemic increased resistance, and the responsiveness of the healthcare systems to outbreaks of communicable diseases, and promoted vaccinations. Many countries addressed improvements in local surveillance, severity estimates, the flexibility of responses, vaccinations, involvement of healthcare workers, and communication [4,5]. The World Health Organization (WHO) recommends a 75% seasonal influenza vaccination rate among elderly people. The European Union Council supports this. However, in Poland, the reluctance toward pandemic vaccinations was extended to the seasonal influenza vaccination; this was observed in subsequent years among all age groups (see Figure 1). The response to the swine flu pandemic had subsequent consequences on the moderate uptake of the COVID-19 vaccine [6]. The annual number of influenza and influenza-like illnesses, parallel to a low vaccination rate, increased systematically from 100,000 cases in the first decade of the 21st century to millions of cases in the past decade [7].

Low vaccination coverage, distrust in the effectiveness of seasonal vaccinations, and false beliefs forced collaborators to change the knowledge, attitudes, trust, and the Polish vaccination organization system. Who ignited this change? They were parents of children negatively affected by influenza and young lawyers who opposed the inert attitudes of the state toward the primary prophylactics and healthcare organizations in Poland. In November 2012, the National Program for Influenza Prevention (NPIP) was established. The NPIP focused its educational initiatives on the medical society, the public, local politics, employers, and finally, the government, to shape the influenza infection prevention and treatment system in Poland. The core of the NPIP was medics; however, the initiative quickly gained support from people in the media, science, culture, and business organizations. In 2022, NPIP had its 10th anniversary, making it one of the oldest educational initiatives in Poland. This review, authored by the members of the NPIP scientific council and supporters, discusses the structure and aims of the program, its operational model, and changes made in the flu vaccination landscape in Poland. We present diversified and long-term activities of the NPIP, which enabled better access to influenza vaccines, modified social attitudes, and gradually increased the vaccination rate in Poland.

## 2. Structure and Aims of the National Program for Influenza Prevention

The NPIP has a transparent structure and principles of governance. The NPIP has a council that maintains and directs initiatives and projects. The council includes leaders of infectious diseases, pulmonology, cardiology, pediatric clinics, national and regional consultants in different disciplines of medicine, representatives of pharmacists, and local authorities. Medical stakeholders proactively shape the organization’s culture to support evidence-based practices in its initiatives. Experts from different regions of Poland support them. The council’s role is to adhere to evidence-based and promising practices supporting a straightforward goal to increase the vaccination rate. Supporting members include healthcare practitioners who help to extend the council’s influence via role modeling, i.e., presenting the behaviors required to carry out strategic actions. Operational changes, e.g., the promotion of vaccinations to peers and patients, are more likely to succeed when professionals become part of the change and recognize its value [8]. Other supporters include lawyers, healthcare system experts, and analytics experts. Scientists require compliance and legal and analytic functions to present the values of the proposed changes and prepare model solutions that decision-makers or practitioners can implement.

Healthcare organizations should use evidence-based practices to improve the quality of care. To fulfill this requirement, data-driven decision-making, combined with practitioners’ knowledge, are key determinants in the performance of NPIP. In recent years, NPIP has striven to create a benchmark for institutionalizing evidence-based practices [9]. Creating solid links between scientific, local, and national political communities was required to integrate scientific evidence into healthcare system decision-making. The NPIP promoted the best practices for a national vaccination strategy by addressing the roles of its different stakeholders: healthcare authorities, practitioners, and the general public.

Vaccinations are the most cost-efficient methods to decrease healthcare costs, morbidity, and mortality of influenza. Considering Poland’s low vaccination coverage, the NPIP focused on identifying actions to increase Poland’s vaccination rate. The VENICE project (completed in 2012) filled the gap in information about national vaccination policies in Europe [10]. Vaccination policies for adults were inconsistent across the European Union and European Economic Area. At the time of the study, in Poland, there were 7 vaccine recommendations for adults our of 19 vaccines analyzed. The study identified four key pillars of national vaccination strategies that drive the prevention of communicable diseases: management of vaccination programs, involvement of healthcare professionals (HCPs), social communication, and access and financing of vaccines/vaccinations. The NPIP considered the results obtained in the VENICE project in their objectives and activities. There were several gaps between Poland and the model vaccination policies in adults, e.g., Poland has fully implemented recommendations for influenza vaccinations in different age groups; however, many other requirements were missing (e.g., a national vaccination threshold, monitoring vaccination coverage in risk groups, incentives to HCPs, ≤90% reimbursement of vaccines), or functioned only partially (e.g., elements of social communication) [11]. Thus, the NPIP aimed to fill these gaps. It started by defining missing elements of the Polish vaccination landscape. The first program document contained a series of recommendations related to critical pillars of a national vaccination strategy [10]. Some of them resonate until today, e.g., increasing the involvement of healthcare institutions to protect vulnerable groups, broadening vaccination reporting and monitoring, increasing HCP’s vaccination coverage, promoting vaccinations, and improving access to vaccinations [11]. The NPIP stimulated the involvement of health authorities, created a platform to join the resources of different stakeholders, increased awareness, educated different target groups, and led the national informational campaign about the flu and vaccinations.

Early, the organization received support from experts from the National Institute of Public Health, the National Influenza Center, the Polish Society of Health Education, the Polish chapter of the European Federation of Pharmaceutical Industries and Associations (EFPIA), and the consultancy company Ernst & Young. They consisted of the core of the future NPIP coalition. Members of the NPIP had a common goal: to increase the awareness of the influenza burden and advocate for the influenza vaccination. In time, the scope of the endorsement of NPIP activities increased by developing links with other groups. In the last 10 years, the organization partnered with the Ministry of Health, the president of Poland, the parliament, the Main Public and Military Sanitary Inspectorates, the Ministry of Labor and Social Policy, employer associations, local authorities, and commercial companies. Public health challenges are complex and can be better solved by stakeholders working together. The diversity of partners helps to address the necessary different aspects of problems, which often go beyond medical issues and require social, economic, and public policy insights. The size and heterogeneity of the coalition indicate that NPIP succeeded in attracting individual members and organizations around the common value.

The organization became the primary source of public information concerning the number of vaccinations performed every flu season. When the official number was limited to the number of vaccines reimbursed by the payer, NPIP combined data from distributors and the National Institute of Public Health to provide the broadest possible view of annual vaccination action (Figure 1). The data presented can guide the orders of vaccines in the next season. Using social science to understand the dynamically changing realities of Poles, as well as their experiences, helps NPIP guide the actions aimed at a higher vaccination uptake. Examples of such initiatives include a series of recommendations for shorting patient paths to vaccinations and the involvement of HCPs in the promotion of seasonal vaccinations. The second document aims to realize that the consequences of the flu go far beyond medical matters. The NPIP prepared a model of the flu epidemic in Poland to calculate the direct and indirect costs of the disease. The indirect costs were 20-times the medical costs, even in a season without an epidemic, and dramatically peaked during the epidemy. The data help researchers understand the burden of the flu from a nonmedical perspective [12].

There are many other examples of evidence-based content and recommendations created by the NPIP. This scientific background is necessary to shape vaccination policies and support the decisions of health authorities. In addition, the fact-based approach easily engages the media—a critical contributor to increase the awareness and education of immunizations.

## 3. Operating Model

### 3.1. Sustainability of the Organization

A change in policy rarely immediately follows a recommendation. The NPIP activities require persistence and patience to pursue objectives over a long period. There was an early reaction to the proposition to establish vaccinations against flus in pharmacies. However, two years later, the Minister of Health was vaccinated by a pharmacist. Thus, resistance to failures and persistence to defined priorities are well established in the NPIP. The work of the NPIP is organized around specific routines that help evaluate effects and plan new actions. At the beginning of each year, the scientific council of the organization meets to plan areas of focus and activities. Despite a shared broad vision, the council focuses the annual agenda of the NPIP on a set of key priorities. They include changes in the healthcare system, roles, processes, capabilities that are required to manage vaccination programs, leveraging HCP’s involvement, and improving social communication and access to vaccines. In recent years, the outcomes of the scientific council meetings were published in the white papers (Table 1), which defined challenges, reviewed accumulated qualitative or quantitative data based on key assumptions and methods, presented model operations and systems, drew conclusions, and stated recommendations. Every time, target groups received suggested next steps with implications of the findings and recommendations.

For example, the white paper on pharmacy-based vaccinations [13] reviewed evidence behind the organizational and administrative feasibility of influenza vaccinations in Polish pharmacies and presented European operating models offering multiple implementation options. Moreover, it analyzed impact on the healthcare system with particular attention to issues related to deficiencies in the healthcare workforce and patient preferences. Since this white paper mainly targeted decision-makers, it reflected their primary concerns and presented the key formal steps necessary to bring vaccinations to pharmacies.

The white paper “Together against infectious disease” targeted HCPs and healthcare institution managers [14]. Here, the NPIP reviewed evidence behind the efficiency of different practices of vaccination promotion and showed that every healthcare practitioner in contact with a patient might have a meaningful impact on vaccination decisions. To identify the most efficient interventions, we explored the libraries of the Cochrane Public Health, the Community Guide, and the International Union for Health Promotion and Education [15,16,17]. Based on this, the NPIP developed vaccination promotion toolboxes for practitioners in different settings and healthcare stakeholders. This supported the role-modeling of multiple HCPs.

Council meeting outcomes and white papers are consult by partners and are released during the annual flu meeting. The past flu season is summarized during the event, considering the incidence of the flu and the vaccination rate. It is also the time to recognize journalists who are involved in driving the change in the flu prophylaxis landscape in Poland. The media is a critical stakeholder in increasing the awareness and education about immunization; thus, incentives to engage with media and recognize good journalism, such as annual health feathers, are apparent.

The time after the flu season is not silent; the distribution of the outcomes continues in the media during the Academy of Flu organized for HCPs and contacts with local authorities implementing health programs/preparing educational campaigns. The new flu season starts with the flu forum, which provides recommendations for the new influenza season, and the experts debate about necessary systemic changes. During the flu forum, representatives of municipalities are awarded for the organization of educational campaigns and local flu prevention programs. The NPIP looks for and takes opportunities to promote influenza vaccinations in different settings. In 2016, before the Olympic Games in Rio de Janeiro, athletes were vaccinated, with vaccines recommended both for northern and southern hemispheres [18]. The competition was the most important in the lives of many participants, so preventing the flu was necessary. However, everyone can do it, not just athletes; it was a central campaign communication supported by the NPIP members.

The NPIP has an annual calendar and rhythm of engagement, making it a leader among national stakeholders. The scope of actions of the NPIP was recognized as the most comprehensive among different vaccination coalitions [19]. This results from the coalition’s diversity in combination with routines and persistence.

### 3.2. Key Actions of the Organization

Every NPIP coalition member has a role to play, and partners work together to leverage their strengths. However, none of the achievements can be attributed solely to an NPIP action since they edge on multiple opportunities and synergies. All actions of NIP are directed to specific groups or individual shareholders defined in the VENICE project [8] and the main program document [9]. In the first ten years of work, the NPIP prepared several reports dealing with essential elements of the national vaccination system (Table 1) [11,13,14,15,16,17,18,19,20,21]. Media and advertisement campaigns supported the dissemination of outcomes and educational content. The effects of actions were often intangible; however difficult they were to measure, they sure contributed to changes in recent years. Polish health authorities repeatedly commented on the actions and recommendations provided directly or in the media. Some actions took place in the parliament directly involving its members. The NPIP became one of the non-governmental organizations invited to consult on changes in the immunization landscape in Poland. These opportunities were used to support healthcare system changes, increasing access to influenza vaccines and vaccinations in Poland (Figure 2).

**Table 1 vaccines-10-02030-t001:** Target group, focus, and main communication topics released by the National Program for Influenza Prevention until 2022.

Target Group	Key Focus	Main Topics	Reference
Healthcare Authorities	Influenza prevention engagement	Personal involvement Recommendations National media campaign	Plan for flu and COVID-19 [20]
	Financing vaccines and vaccinations	Reimbursement Financing vaccinations Flu in NIP	Flu and its costs [12] Plan for flu and COVID-19 [20]
	Systemic changes	Vaccination competences Pharmacy-based vaccination Data-driven vaccination planning	Vaccinations in pharmacies [21] How to vaccinate in a pharmacy [22] Plan for flu and COVID-19 [20]
Healthcare Practitioners	Recommending vaccinations to patients	Rationale, motivation and incentives	Strength of HCP recommendation [14] How to recommend vaccination in risk groups [23]
	Self-vaccination	Self-awareness, incentives and obligation	Vaccinations of HCPs [24]
	Flu awareness	Awareness of flu consequences	Safe Medical Facility [25]
Public	Flu prevention in local health programs	Planning flu vaccination programs	Model program of flu prevention [26,27] Plan for flu and COVID-19 [20] Epidemiologic calculator
	Vaccinations in public services staff	Engagement of the Ministry of Defense and Ministry of Internal Affairs and Administration	Plan for flu and COVID-19 [20]
	Vaccination of employees	Engagement of the Ministry of Labor and employer’s organizations	Plan for flu and COVID-19 [20] Employers at risk of the flu [28] Flu and its costs [12]
	General Public	National campaign and regular events	Flu-resistant [29] Facts and myths about the flu [30]

HCP, healthcare practitioners; NIP, National Immunization Program.

Key initiatives targeting health authorities were to increase the influenza vaccination in the hierarchy of obligatory and recommended vaccinations in the annual National Immunization Program (NIP), broadening access to vaccinations through reimbursement, and creating possibilities to get vaccinated. The NPIP formed an additional coalition for vaccinations in pharmacies with the Main Chamber of Pharmacy in Poland, supported by members of the Pharmaceutical Group of the European Union (PGEU) and the Employed Community Pharmacist in Europe (EPhEU). The new coalition collected evidence and showed that vaccinations at pharmacies are safe and desired by society [21,31]. This was confirmed by a pharmacist’s readiness and patient satisfaction checks [32,33]. The COVID-19 pandemic accelerated the development of competencies to qualify for vaccinations in the case of other HCPs in Poland. This opened up new opportunities, e.g., vaccinations can easier leave healthcare facilities and step into nonmedical settings, e.g., schools. This is a critical need; the path for children’s vaccinations is the longest in the system, and the vaccination rate remains low despite the recent increase in other age groups (Figure 1) [20].

The NPIP reviewed the different strategies for promoting vaccinations and concluded that these (based on HCP recommendations) are the most effective. Every profession had an outline of activities [14] and practical examples [23] on how they could increase vaccination coverage. Health authorities developing the National Plan of Immunization against COVID-19, and were informed about the analysis outcomes and the value of professional vaccination recommendations.

The NPIP collaborates with local authorities regularly. In 2014, the NPIP and the Central and Eastern European Society of Technology Assessment in Healthcare prepared guidelines, a calculator, and a model of the flu prevention program for local governments willing to invest in prevention [27]. They consisted of a toolkit to prepare the program according to standards required by the Polish Health Technology Assessment Agency. Thus, local authorities could limit the administrative burden of preparation of independent programs. The NPIP gives the rationale behind vaccinations in different professional settings. The flu economically impacts employers and endangers work continuity. The operations of companies can be protected with annual vaccinations of employees. The possibility of receiving a flu shot at work became a tradition in many enterprises in Poland. The above-mentioned activities increase the number of opportunities to get vaccinated and are supported by the promotion of vaccinations to the general public.

## 4. Outcomes

The complexity of the vaccination landscape makes it impossible to attribute success to a single stakeholder, even when the NPIP operates based on a broad coalition. The NPIP contributed to policy changes, guidelines, laws, and attitudes toward vaccinations in the last ten years. In 2014, influenza vaccines began to be recommended in the NIP to all children above six months as well as adults instead of only to the groups with high risks of the disease (Figure 2). At that time, vaccines were not reimbursed, consisting of out-of-pocket costs in most cases. This changed in 2018. Older adults (≥65 years old) received 50% reimbursement of the flu vaccine cost. In 2020, they were accompanied by the following groups: children from 2 to 5 years old and adults at risk of severe illness. Persons ≥75 years old received a 100% reimbursement of the cost of vaccines. Access to reimbursed vaccines in pharmacies and the development of pharmaceutical care services created an environment to propose vaccinations in a pharmacy setting. The NPIP drove the evolution of vaccination competencies. Arguments about the decreased workloads of physicians and nurses resonated at the time of the COVID-19 pandemic. Pharmacists and other professionals, after training, had the right to qualify for vaccinations, which was the sole responsibility of physicians (Figure 2). Around 1 million Poles were vaccinated against COVID-19 in 2021, and new competencies were broadened to vaccinate against the flu. The NPIP supported legal and educational changes required to launch a new pharmaceutical service [21]. Nowadays, the NPIP advocates for subsequent changes, including the possibility of issuing a prescription for a vaccine by pharmacists, or not requiring it for vaccines administrated in pharmacies [20].

The COVID-19 pandemic changed the vaccination landscape in Poland. Vaccinations have never been so crucial for adults before. In the 2021/2022 season, flu vaccines were provided for free for the first time to all adults. Influenza vaccination coverage in older adults reached over 22% as this group shares a common risk of severe courses of both diseases. The NPIP recommended the joint plan for flu and COVID-19, which is based on the synergies between both vaccinations and aims for further improvements in the vaccination coverage [20].

## 5. Summary

After a dip and years of stabilization at low levels, vaccination coverage started to grow again in recent years (Figure 1). Hopefully, Poland will soon achieve a vaccination rate similar to that before the A/H1N1 flu pandemic. Vaccination coverage among older adults is historically high, but it remains low in younger age groups. This will become the main focus of the NPIP’s actions in the future. The COVID-19 pandemic proved again that vaccines are among the most outstanding achievements of medicine. Shaping access and the attitudes of the youngest people toward vaccinations will lead to higher vaccination rates. Although the vaccination rate has increased, it is still very low, which indicates the need for ongoing educational campaigns aimed at both patients and doctors to raise the awareness of the benefits of vaccinations. Upcoming influenza seasons may have substantial infection burdens across the globe. Timely influenza vaccination programs would be the best preventive measures to reduce the impact of influenza in the community. The current recommendations for the organization of targeted population vaccination programs, school-based vaccination promotion, and joint vaccinations against COVID-19 and influenza [34] align with those formulated by the NPIP in early 2022 [22]. The NPIP will continue to increase the vaccination rate in the following years.

This review showed that the long-term and diversified activities bring the desired changes in providing access to vaccines (e.g., reimbursement) and vaccinations (e.g., pharmacy-based vaccinations), as well as a gradual increase in the percentage of Poles vaccinated against influenza.

## Figures and Tables

**Figure 1 vaccines-10-02030-f001:**
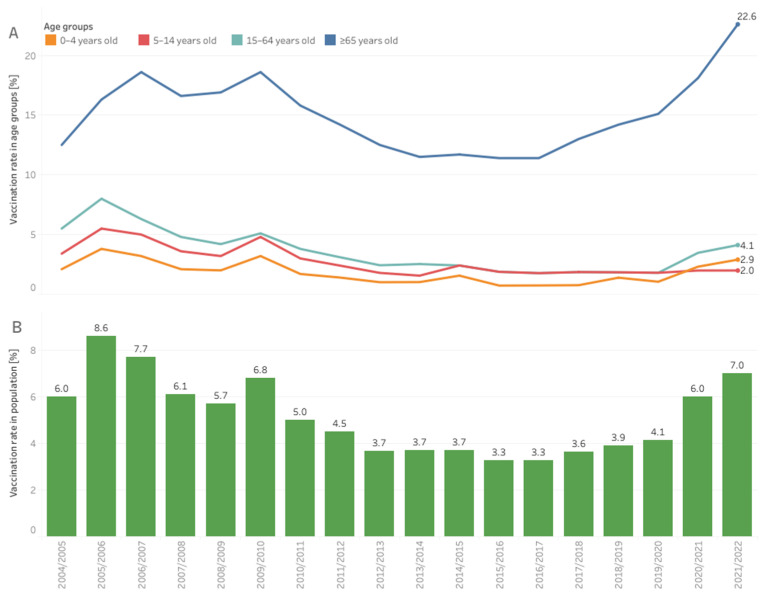
Influenza vaccination coverage in Poland in individual age groups, (**A**) the entire population, (**B**) in influenza seasons from the beginning of the XXI century. Market estimates based on data from distributors and the National Institute of Public Health.

**Figure 2 vaccines-10-02030-f002:**
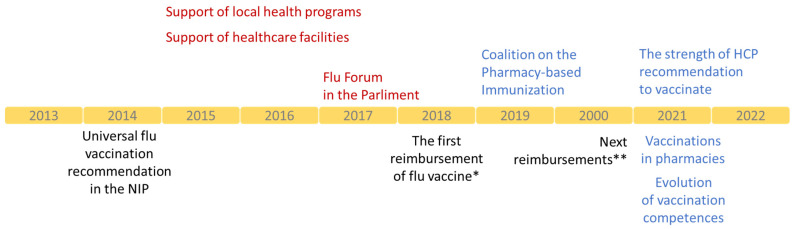
Main actions of the NPIP and milestones of evolution of influenza vaccination in Poland, *****-50% reimbursement of influenza vaccine for patients aged 65–74 years old and 100% reimbursement for patients ≥75 years old; **-50% reimbursement of influenza vaccine for patients aged 18–64 years old from risk groups, pregnant women, and children aged 2–5 years old.

## Data Availability

Not applicable.

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
