# Peer review of "Coalition Shaping the Vaccination Landscape"

_vaccines, 2022, doi:10.3390/vaccines10122030_

Round 1
Reviewer 1 Report
The introduction is very long and argumentative. Often it is difficult to follow the exact themes of paragraphs. This could be reduced in length and focused on the nature of the data and specific reasons for the report.
Reading the manuscript was confusing; I was not sure if this was an historical review or a presentation of new information.
I would suggest the authors review and revise the approach for the paper. Also, extensive issues were observed with word choice and sentence structure.
This manuscript requires extensive revision and probably re-writing. However, the limited data presented are interesting.
Author Response
Dear Reviewer,
Thank you for your review. We did not reduce the manuscript length since the Editorial Office requested to extend it. For clarity, in the introduction, we stated the aim of the manuscript: to present diversified and long-term activities of the NPIP, which enabled better access to influenza vaccines, modified social attitudes, and gradually increased the vaccination rate in Poland. We introduced the organization's structure, operational model, significant activities, and outcomes in the manuscript. Thus, we agree with the conclusion that it is a historical review. We revised the English and improved readability of the manuscript. We changed the order of some paragraphs.
Other significant changes include:
- Extension of part describing the roles of the scientific council and types and roles of supportive members.
- Added multiple elements helping to evaluate the evidence-based practice of the organization from the perspective of benchmarks characterized by Stetler CB et al., Implementation Science 2009, 4: 78.
- Explained methods and design of two examples of white papers issued by the organization; one directed to healthcare decision-makers and one for HCPs and healthcare institution managers.
Kindest Regards,
Authors of the manuscript entitled "Evidence-based initiative shaping vaccination landscape"
Reviewer 2 Report
The article presents a discussion on the activities carried out by a National Program for Influenza Prevention (NPIP) in Poland since 2012. The interest of the paper is very limited, as it not a scientific review of the activities carried out but a relate about the NPIP without scientific merit. I suggest to the authors to send the article to another type of journal, less focused on the vaccines and vaccination research.
The article has the following problems due to the type of relate used by the authors to discuss about the NPIP:
1. The structure and function of the coalition called NPIP is not explained in sufficient detail (line 187). It is not clear what are the roles/activities of the authors of the paper in the NPIP, as the authors does not present any document about this.
2. The article does not follow the methodology explained by Steler et al. (Implementation Science 2009; 4:78) for discussing about an institutional evidence-based practice. For this reason, one of the key weakness of the article is that it is not possible to know the scientific merit of the activities developed by the NPIP
3. The relation between the VENICE project and the activities of the NPIP (lines 190-193) is not explained in sufficient detail. The objective of the VENICE project was to fill the gap of information regarding vaccination policies in adults in countries of the European Union (EU) and Economic Area (EEA). The conclusion of the project was that vaccination policies for adults are not consistent across Europe. The paper however does not explain whether the NPIP had taken into account the results obtained in the VENICE project in their objectives and activities.
4. The article does not explain in sufficient detail how the activities of the NPIP were implemented, including the objectives, available resources, activities carried out, and results obtained. For example, the vaccination of athletes related to the reference 12, explained on lines 176-179, was carried out for assessing the effects of influenza vaccines without the participation of the NPIP. The NPIP was invited to consult on changes in immunization sin Poland (line 199), but there is not any comment about how this activity was carried out, what were the results, and the documents derived from this activity.
5. The article indicates that a number of reports (references 13 to 21) have been developed by the NPIP (line 193). Nevertheless, the paper does not explain the content of these reports, how they were developed, the participation of the authors, the communication strategy, the resources used and the results obtained in terms of influenza vaccination promotion.
Author Response
Dear Reviewer,
Thank you for your review. Our manuscript is intended to be published in the section Vaccines and Society of the Vaccines journal. The manuscript does not focus on the scientific aspects of vaccine research but discusses institutional efforts to increase access to vaccines and vaccinations. However, vaccines have changed medicine; vaccinations are necessary to bring these advantages to the practice. We believe that institutional efforts are necessary to shape the attitudes of decision-makers, healthcare practitioners, and the general population toward vaccinations. In the introduction, we explained reasons behind negative attitudes toward recommended vaccinations in Poland. Since these frames of thinking about vaccines and vaccinations were prevalent among politics, practitioners, and society, an educational organization was established to diversify a long-term approach to change the influenza vaccination landscape in Poland. This manuscript summarizes the organization's structure, operational model, history of activities, and outcomes associated with changes in the vaccination landscape in Poland that happened in the last ten years.
We addressed points requiring additional explanation in the following way:
- The authors of the paper are members of the scientific council of the NPIP and supporters, and we explained the roles of every author in the Acknowledgments. In addition, we added additional information about the roles of the scientific council and supporters (tracked changes in the section Structure and aims of the National Program for Influenza Prevention
- We agree that the manuscript does not follow methods described by Steler et al., 2009 to evaluate a degree of evidence-based practice in healthcare organizations. Since these involve many dimensions of functioning (internal, response to external pressure, leadership, organizational structure, decision-making) we added new parts documenting the functioning of the NPIP and adhering to best evidence-based practices. Using two examples, we described how the NPIP creates white papers, one targetting healthcare decision-makers and one for HCPs and healthcare institution managers. All changes were tracked.
- We mentioned that the VENICE project outcomes shaped actions of the NPIP. The main gaps identified in the VENICE project, which were also addressed by the NPIP, were the management of vaccination programs (e.g., vaccinations in pharmacies, school-based programs, health programs developed by local authorities), healthcare professional's involvement (vaccinations of healthcare practitioners, how to recommend vaccination to a patient), social communication (general public awareness campaign, workplace vaccinations), and access and financing of vaccines/vaccinations (reimbursement of Vaccines, reimbursement of vaccinations in pharmacies). We modified part about the VENICE project, and changes are tracked.
- We added explanations to mentioned activities. The vaccination of athletes was carried out with the support of NPIP, and members of the scientific council of the NPIP were involved in the project. Please find that the authors of the cited manuscript are also the authors of the current paper. We confirm that the NPIP has become one of the organizations regularly consulted about regulations associated with preventing communicable diseases. It involves requests to join consultations (e.g., participate in works of parliament commissions) and proactive statements.
- We presented how white papers are developed in the NPIP and used examples of two recent documents. These new parts are tracked.
Kindest Regards,
Authors of the manuscript entitled "Evidence-based initiative shaping vaccination landscape"
Round 2
Reviewer 1 Report
The manuscript has been improved and is ready for final editing at the discretion of the journal editors.
Author Response
Thank you.
Reviewer 2 Report
The article has the following problems:
1. The structure and function of the coalition called NPIP is not explained in sufficient detail, and the authors does not include documents about the NPIP, including why was developed, the roles of their members, theur detailed activities and how they have been evaluated.
2. The article does not follow the methodology explained by Steler et al. (Implementation Science 2009; 4:78), and the authors do not explain in a suficiently consistent way its scientific value and why it can be of interest from an international point of view.
3. The relation between VENICE project and the activities of the NPIP is not explained in sufficient detail, as it does not include concrete interventions of tthe NPIP and their results.
4. The article does not explain in sufficient detail how the activities of the NPIP were implemented, including the objectives, available resources, activities carried out, and results obtained.
5. The relation between several of the papers commented in the article and the NPIP is still not sufficinetly clear. For this reason, the article is a comment about works carried out in Poland related to infectious diseases, more than about an evidence-based initiative.
Author Response
Thank you for your comments. We decided not to change the manuscript. We changed the title, to not adhere to an evidence-based initiative but simply to a coalition. We confirm that the article discusses our organization's history, background, and aspirations. We will not evaluate our compliance with the standards of evidence-based organizations since we are not legitimate to do this. We are striving to these standards. We cannot tell that changes in the vaccination landscape in Poland are only our merit, and we clearly stated that in the manuscript. However, we feel the NPIP was a part of these positive changes, i.e., we actively educated and solicited them. We did not review each of our activity year by year, but we presented a routine framework that allowed us to address regularly different elements of national vaccination strategy, i.e., management of vaccination programs, involvement of healthcare professionals (HCPs), social communication, access and financing of vaccines/vaccinations. These were the main pillars of strategies identified in the VENICE project, and we built on its foundation.
Round 3
Reviewer 2 Report
The title reflects the content of the article. Despite the explanations on the structure and role of the NPIP coalition members is very limited, as they are comented on the "authors contributions" statment, the text of the article is a good story.